# WHO SAID THAT? BENCHMARKING SOCIAL MEDIA AI DETECTION

## ABSTRACT

AI-generated text has proliferated across various online platforms, offering both transformative prospects and posing significant risks related to misinformation and manipulation. Addressing these challenges, this paper introduces SAID (Social media AI Detection), a novel benchmark developed to assess AI-text detection models' capabilities in real social media platforms. It incorporates real AI-generate text from popular social media platforms like Zhihu and Quora. Unlike existing benchmarks, SAID deals with content that reflects the sophisticated strategies employed by real AI users on the Internet which may evade detection or gain visibility, providing a more realistic and challenging evaluation landscape. A notable finding of our study, based on the Zhihu dataset, reveals that annotators can distinguish between AI-generated and human-generated texts with an average accuracy rate of 96.5%. This finding necessitates a re-evaluation of human capability in recognizing AI-generated text in today's widely AI-influenced environment. Furthermore, we present a new user-oriented AI-text detection challenge focusing on the practicality and effectiveness of identifying AI-generated text based on user information and multiple responses. The experimental results demonstrate that conducting detection tasks on actual social media platforms proves to be more challenging compared to traditional simulated AI-text detection, resulting in a decreased accuracy. On the other hand, user-oriented AI-generated text detection significantly improve the accuracy of detection.

## 1 INTRODUCTION

The advent of AI-generated text has had a profound impact on numerous sectors, including social media platforms. On one side, AI-generated responses enable automation, personalization, and scaling of content creation, thereby revolutionizing how information is disseminated and consumed. However, this promising technology has a darker aspect, where it is exploited for disseminating misinformation, impersonating real users, or creating content that can deceive or manipulate public opinion Ji et al. (2023); Chen et al. (2023); Li et al. (2023); Lin et al. (2022); Lukas et al. (2023); Perez et al. (2022); Zhuo et al. (2023); Santurkar et al. (2023).

Addressing the abuse and malicious use of AI has led to significant research efforts in the field of AI-generated text detection. A range of approaches have been explored, including but not limited to machine learning algorithms Guo et al. (2023); Solaiman et al. (2019), text-based features analysis Mitchell et al. (2023); Tulchinskii et al. (2023); Mitchell et al. (2023), and positive unlabeled techniques Tian et al. (2023). These endeavours aim to automatically discern machine-generated text from human-generated content, thereby mitigating the risks associated with its abuse.

One of the most significant shortcomings of the existing research landscape is the lack of benchmarks that accurately represent the complexity and variety of AI-generated text in real-world scenarios. Many of the existing benchmarks employ artificial, controlled environments, such as directly querying LLMs for generating AI content Mitchell et al. (2023); Guo et al. (2023); Krishna et al. (2023). While such settings may be useful for preliminary evaluation, they often fail to capture the nuanced tactics that are deployed in the wild to evade detection or maximize exposure. For example, AI-generated text posted on social media may be crafted using a combination of strategies like text modification Cai & Cui (2023) or paraphrasing Krishna et al. (2023), thereby making it harder for traditional detection methods to identify them.

Table 1: Statistics of human and AI-generated responses in SAID.

|  | SAID-Quora | | SAID-Zhihu | |
|---|---|---|---|---|
|  | Human | AI | Human | AI |
| Users | 491 | 464 | 6769 | 4565 |
| Responses | 14669 | 22902 | 72568 | 108654 |

On the other hand, existing datasets for AI-generated text on realistic scenarios were assembled before 2022, the extensive deployment of LLM (e.g. TweepFake-2021 Fagni et al. (2021), XSum-2018 Narayan et al. (2018), Kaggle FakeNews-2018 Lifferth (2018)). The relevance and efficacy of these datasets are now questionable for evaluating detection capabilities against contemporary, more advanced LLM-generated text.

In light of these limitations, we introduce SAID (Social media AI Detection) — a novel benchmark aimed at addressing the gaps in existing AI-generated text detection benchmarks. SAID is unique in its approach of crawling real AI-generated data from popular social media platforms, such as Zhihu.com and Quora.com, to provide a more accurate and challenging assessment of the current detection models. We aim to advance the field of AI-generated text detection by offering a more realistic testing ground.

One significant challenge in collecting real machine-generated text is determining the ground truth of whether a piece of text is AI-generated. Fortunately, we found that Zhihu itself provides automated detection labels for AI responses, serving as a foundation for our AI labeling. Initially, we consider responses with this label as AI-generated. Subsequently, we extend this label to a user level. In this paper, our major assumption is that **users with AI-generated responses are likely to be an AI, and therefore their other responses are also AI-generated**.

A prevailing notion previously posited by OpenAI suggests that humans are incapable of distinguishing between AI-generated and human-generated texts Solaiman et al. (2019). However, from the SAID-Zhihu, one of the vital conclusions of this paper is that **annotators, adept with LLMs and social media, can accurately distinguish whether a response is AI-generated**, achieving an average recognition accuracy rate of up to 96.5%. We do not perceive this conclusion as entirely contradictory to the findings of Solaiman et al. (2019), but rather reflective of a shift in context. In 2019, when the conclusion of OpenAI was drawn, AI-generated text was in its infancy and not widely applied. Humans had a limited understanding of AI-generated text at this time, thus, smooth and coherent text on the internet was generally believed to be human-generated. However, by 2023, when our experiments were conducted, AI-generated text had matured and yielded societal benefits. Hence, individuals familiar with AI-generated text, having encountered ample examples, have developed the capability to distinguish it from human-generated content. Consequently, we argue that in today's era, where AI-generated content widely impacts individuals, it is crucial to re-establish the understanding of humans' ability to recognize AI-generated text.

Armed with this new insight, we proceeded to label AI-generated and human-generated responses on Quora. To enhance the efficiency of extracting AI-generated text, we collected AI-generated responses from the *collapsed* answers. Statistics of SAID are presented in Table 1.

We then evaluated the performance of current AI-generated text detectors, which are mostly developed based on simulated data.The results indicate that, when tested on the real SAID dataset, the performance of detectors experienced a noticeable decline compared to previous simulated data, such as HC3 Guo et al. (2023). Consequently, SAID presents new challenges for developing genuinely practical AI-generated text generators, emphasizing the necessity of improving the reliability and robustness of detection methods in real-world applications.

Moreover, we introduce a novel and practical challenge: user-oriented AI-generated text detection, identifying whether a response is AI-generated or human-generated based not only on its content, but also on its user information like the user's other responses. This task proves to be more practical and effective for AI-generated text detection. As AI users tend to post voluminous AI-generated text, the other answers from the user provide abundant valuable information for the detection, thus significantly enhancing the detection accuracy. Experimental results substantiate our assertion.

In summary, the main contributions of our work are as follows:

1. **Novel Benchmark for AI-generated text Detection:** We introduce SAID, a benchmark offering real-world AI-generated text from social media platforms, Zhihu and Quora. This addresses the shortcomings of existing benchmarks by providing more realistic and diversified AI detection.

2. **Insightful Findings on Human Discernment:** Our work unveils the noteworthy finding that, contrary to prior assumptions, individuals familiar with LLMs and social media can accurately distinguish between AI-generated and human-generated responses, achieving an impressive 96.5% average accuracy. This prompts a re-examination of human cognition capabilities in AI-generated text detection in the modern AI-infused environment.

3. **User-Oriented AI-Generated Text Detection Challenge:** We propose and illustrate a novel user-oriented AI-generated text detection challenge. This emphasizes the importance and effectiveness of leveraging user information and content analysis for more practical and robust AI detection. We demonstrate its effect through experimental validation.

4. **Empirical Evaluation of Current AI Detectors:** By deploying contemporary AI-generated text detectors on SAID, we present a detailed empirical evaluation, uncovering their capabilities and limitations in real-world scenarios. We found clear performance decline in SAID. This offers novel insights and directions for enhancing the reliability of AI detection methods.

## 2 RELATED WORK

The burgeoning field of AI-generated text detection has witnessed numerous contributions, exploring diverse detection strategies and methodologies. Here, we elaborate on the pertinent works and the evolution of AI-generated content detection, illustrating the context in which our research is positioned.

**AI-Generated Text Detection Approaches** Various studies have introduced innovative solutions to tackle the challenges posed by AI-generated text, using machine learning algorithms Guo et al. (2023); Solaiman et al. (2019). These approaches extensively analyze textual features to differentiate between human and AI-generated content Mitchell et al. (2023); Tulchinskii et al. (2023). Guo et al. (2023) and Solaiman et al. (2019) have played a pivotal role in shaping the early development of detection algorithms, leveraging classical machine learning models to discern the nuances in AI-generated text structures. As explored by Tian et al. (2023), positive unlabeled techniques have surfaced as promising avenues for enhancing detection accuracies. These techniques effectively deal with the scarcity of labeled AI-generated text instances in real-world scenarios by utilizing the available positive and unlabeled samples to train robust models.

**Feature Analysis** The focus on text-based feature analysis is evident in the works of Mitchell et al. (2023) and Tulchinskii et al. (2023). These studies delved into the inherent features and patterns of texts to devise effective detection mechanisms, exploiting the subtle discrepancies between human and machine writings. The intrinsic analysis of text has proven vital in unveiling the hidden characteristics and intricacies of AI-generated content.

**Evasion Tactics and Paraphrasing** The increasing sophistication in evasion tactics is highlighted in studies such as Cai & Cui (2023); Sadasivan et al. (2023); Krishna et al. (2023). These works emphasize the escalating challenges posed by advanced text modification and paraphrasing techniques, which mask the identifiable traits of AI-generated text, necessitating the development of more resilient detection methodologies.

**Benchmark Datasets** Existing benchmarks have largely been constrained by artificial and controlled environments, limiting the representative scope of AI-generated content in real-world conditions Mitchell et al. (2023); Guo et al. (2023); Krishna et al. (2023). Fagni et al. (2021); Lifferth (2018); Narayan et al. (2018) documented datasets compiled before the proliferation of large language models (LLMs), thus questioning their relevance and efficacy against modern, more advanced AI-generated text.

## 3 COLLECTING SAID-ZHIHU

In this section, we detail the methodology deployed to collect SAID-Zhihu.

### 3.1 INITIAL POOL OF AI-GENERATED TEXT

The major challenge in constructing a realistic dataset for AI-generated text detection in social media is obtaining groundtruth labels indicating whether content is AI-generated. On one hand, several social media platforms impose stringent policies restricting AI-generated content, exemplified by StackOverflow's prohibition of AI responses. On the other hand, AI users often employ strategies to make their responses appear more human-like.

Fortunately, Zhihu itself provides labels for a part of AI-generated responses. While Zhihu does automatically detect and limit the display of AI responses—collapsing detected AI responses and labeling them as "Suspected AI Creation"—such responses remain accessible and expandable. Users suspected to be AI can still post normally on Zhihu. We, therefore, use responses labeled as "Suspected AI Creation" as our initial pool of AI-generated texts.

### 3.2 AI-GENERATED TEXT EXPANSION STRATEGY BASED ON AI USERS

Only collecting the labeled responses as AI-generated is not enough. A drawback of the above initial pool is its limited coverage; it only comprises a portion of the AI-generated texts on Zhihu. Relying solely on this pool for AI detection benchmarking has limited significance, as it doesn't assess the detection capabilities beyond what Zhihu detects.

To overcome this, we observed that when a user has posted AI-generated responses, it is highly likely that the user is an AI. Hence, we hypothesize that other responses from this user, although possibly not labeled by Zhihu, are also AI-generated. Following this intuition, we propose the hypothesis below:

**Hypothesis 1** *If an user has some AI-generated responses, then other responses of the user also tend to be AI-generated.*

In this paper, we transform this hypothesis into a concrete label expansion strategy: if more than three out of the first ten responses from a user are labeled as "Suspected AI Creation," we consider this user as an AI and all its responses as AI-generated.

### 3.3 FILTERING

To enhance the confidence in the labels of the collected responses, we employ two filtering strategies for collected AI-generated responses:

- We only consider responses with a character length greater than 250. This criterion aligns with our observation that establishing whether short responses are AI-generated is inherently challenging. Several AI detectors, like GPTZero, align with similar filtering strategy, supporting the detection of texts longer than a specific threshold.

- We limit our consideration to responses posted after March 2023. The emergence of Chat-GPT at the end of 2022 marks a milestone event for AI-generated texts. We posit March 2023 as a temporal node when a substantial number of users began to use ChatGPT for generating content.

### 3.4 RESULTS

Table 1 presents the statistical data of our collected SAID-Zhihu dataset. For comparison, we also collected comparable number of human-generated responses. Here, we consider users whose answers do not carry the "Suspected AI Creation" label as humans.

Table 2 provides an example from SAID-Zhihu. It is observable that the AI-generated responses in it exhibit distinct AI characteristics—they are more vague, less specific, and more formalized.

Table 2: Examples of AI and Human-generated Responses from SAID-Zhihu and SAID-Quora. Texts in green indicate features participants thought are critical in AI-generated text detection.

| |
|---|
| **Zhihu-AI** |
| Is it possible to successfully pass the graduate entrance exam for control engineering within six months? |
| **Response:** Firstly, I want to remind you that control engineering is a highly complex discipline that requires a strong foundation in mathematics and abstract thinking abilities . . . |
| Secondly, gaining admission . . . |
| Lastly, first-tier universities . . . |
| In summary, if you are well-prepared and genuinely passionate about control engineering as a discipline, it is possible . . . |
| **Zhihu-Human** |
| What subtle hints did you miss initially? |
| **Response:** During the summer vacation before starting the second year of junior high school, one afternoon, it drizzled outside. The girl I had secretly admired for a long time came to ask me for help with a computer issue. On that rainy day, we both craved spicy hot pot, and conveniently, there was a hot pot place located between my house and hers. I said, "I want to eat too!" She asked if I wanted to go buy some, but it was raining outside. I said it was no problem, a little rain wouldn't bother me even without an umbrella. I'll go, you just wait at home. . . . |
| **Quora-AI** |
| **Question:** Which algorithm is best suitable for studying protein folding? |
| **Response:** "The study of protein folding is . . . Here are a few notable examples: |
| 1. Molecular Dynamics (MD): Molecular Dynamics simulations . . . |
| 2. Monte Carlo (MC) Methods: Monte Carlo methods use . . . |
| . . . |
| It's worth noting that different algorithms and approaches have their own strengths, limitations, and areas of applicability. |
| **Quora-Human** |
| **Question:** Does light come out of stars at an infinite number of angles? Is that why two people standing next to each other are able to see the same star thousands of light years away? |
| **Response:** Well, there aren't an infinite number of photons |
| But that finite number of photons are spread randomly around a continuous sphere, not fixed to discrete angles. |
| So yes, that's why light from the same star can reach both you and another person, no matter how tiny the angle between you. |

## 4 CAN HUMANS DIFFERENTIATE BETWEEN AI-GENERATED AND HUMAN-GENERATED TEXTS?

**Finding 1** *Humans distinguishes AI-generated and human-generated responses with high accuracy.*

Previous research Solaiman et al. (2019) by OpenAI posits that humans struggle to distinguish between AI-generated and human-generated texts. However, as illustrated in Table 2, there are discernable differences between AI-generated and human-generated content. Consequently, in this section, we endeavor to investigate, using SAID-Zhihu as our research subject, whether humans can effectively distinguish between AI-generated and human-generated text.

**Setup** We randomly selected 100 AI-generated and 100 human-generated responses from SAID-Zhihu. These responses were organized into 100 pairs, each containing one response of each label. The orders of the two responses were randomized. Two participants, familiar with LLMs such as ChatGPT and social media, were involved in the tests. The participants were tasked with identifying which response in each pair was AI-generated, with the evaluative metric being the accuracy of the participants. Random guessing would result in an accuracy of 50%.

The results are presented in Table 3. Surprisingly, the results contrast with the results reported in Solaiman et al. (2019). While early experiments Solaiman et al. (2019) reported an accuracy of

Table 3: Accuracy of participants in identifying AI-generated text.

|  | Participant 1 | Participant 2 | Average |
|---|---|---|---|
| Accuracy | 99 | 94 | 96.5 |

Table 4: Common reasons for identifying text as AI-generated and their frequency. Here a / b indicates the frequency in AI-generated responses and in human-generated responses, respectively.

| Reason | Description | Quora | Zhihu |
|---|---|---|---|
| Enumerations | Usage of enumerative structures (*1/2/3/4/5*, *firstly/secondly/lastly*) or bulleted contents. | 74%/5% | 76%/18% |
| Conclude at the end | Presence of a summarizing statement at the end. | 44%/5% | 62%/9% |
| Specific Phrases | Employment of specific phrases such as *it's worth noting*, *please note*, *remember*. | 65%/4% | 79%/9% |
| Too objectivity | The response is overly objective and lacks subjective opinions. | 93%/16% | 90%/14% |
| Formal style | The style of the reply is overly formal. | 83%/11% | 88%/11% |

54%, the participants identified the categories of the responses with an average accuracy of 96.5%. We attribute this to a changing landscape: starting from the end of 2022, as more people began utilizing LLMs to generate texts, they became increasingly acquainted with the characteristics of AI-generated text. As a result, individuals have developed the capability to differentiate between it and human-generated text.

The results from Table 3 also suggest hypothesis 1 holds true, and the user-based expansion strategy set forth in Section 3.2 is effective. This is because the responses expanded based on this strategy could only be identified with such high accuracy if the expanded responses by AI users have consistent features as the initial AI responses.

### 4.1 How Humans Distinguish: Insights into Human Differentiation Strategies

To further delve into how humans make distinctions between AI-generated and human-generated texts, we asked the participants to elucidate the reasons they identified the texts as AI-generated. We list the five most frequently mentioned reasons along with their proportions in Table 4.

It is evident that current AI-generated texts still possess explicit features from human-generated texts. This reinforces the notion that humans are indeed capable of differentiating between AI-generated and human-generated texts.

## 5 Collecting SAID-Quora

### 5.1 Data Collection

Similar to Zhihu, Quora implements an answer-collapsing policy where responses of low quality are collapsed. These responses are labeled with the reason for collapsing. We noticed that AI-generated texts are also regarded as low quality and collapsed by Quora. Hence, we, analogous to our strategy with Zhihu, procured AI-generated responses from the collapsed answers.

However, unlike Zhihu, Quora does not provide a specific "Suspected AI-generated" tag in the reasons for collapsing. Therefore, based on Finding 1, we employed participants to further annotate whether the collapsed answers are AI-generated. To improve annotation accuracy, in line with Hypothesis 1 and the finding we will present in Section 7, we instructed the annotators to label at the user level, i.e., to determine whether the user of a collapsed answer is an AI, rather than labeling individual responses. If an user is labeled as AI, we regard all its responses as AI-generated. We collect AI-generated responses using this strategy.

For comparison, we also collect human-generated responses. We randomly selected responses and their users from the pool of uncollapsed answers, considering them as human-generated. We discovered that a non-negligible proportion of uncollapsed answers and their corresponding users on Quora were AI. Hence, we also required annotators to label whether the users corresponding to uncollapsed answers were AI.

We present the statistical results of the collected data from Quora in Table 1 and illustrate examples of AI-generated and human-generated responses on Quora in Table 2. It is discernible that responses on Quora, whether AI-generated or human-generated, still have distinctive features as on Zhihu.

## 5.2 HUMAN CAN DISTINGUISH RESPONSES ON ZHIHU. CAN THEY GENERALIZE SUCH ABILITY TO QUORA?

**Finding 2** *The features outlined in Table 4 are effective for both Quora and Zhihu.*

One major concern of the collection strategy for SAID-Quora is, whether Finding 1 from the Chinese Zhihu, can be generalized to the English platform Quora. Only if this holds true, participants can differentiate AI-generated responses in Quora. We posit that such generalization is plausible due to the following reasons: (1) The features used by participants in Table 4 are language-independent; (2) A significant proportion of responses annotated as AI-generated on Quora also align with these features. We demonstrate the coverage ratio of each feature in Table 4. It is evident that these features continue to aid participants in distinguishing AI-generated responses from human-generated ones substantially.

## 6 EVALUATION OF STATE-OF-THE-ART AI DETECTORS ON SAID

In this section, we evaluate the performance of three AI detectors—GPTZero (Tian, 2022), HelloSimpleAI (Guo et al., 2023), and MPU (Tian et al., 2023)—on the SAID dataset. The aim is to understand their effectiveness in distinguishing between AI-generated and human-generated texts on real-world social media.

### 6.1 TEST SETUP

For the evaluation, we randomly selected 100 AI users and 100 human users from each split of the SAID dataset. The detectors were deployed to classify the label of their responses, leveraging their pre-existing models. The metrics for assessment include accuracy, complemented by precision, recall, and F1-score to provide a holistic view of each detector's performance.

We consider the following three detectors:

- **GPTZero** (Tian, 2022) We observed that GPTZero is incapable of accurately detecting text in Chinese. Consequently, we confined the use of GPTZero to the testing of SAID-Quora only.
- **HelloSimpleAI** (Guo et al., 2023) For our experiments on SAID-Quora and SAID-Zhihu, we deployed its English and Chinese versions, respectively.
- **MPU** (Tian et al., 2023) We applied its English-adapted version for SAID-Quora and Chinese-adapted version for SAID-Zhihu.

### 6.2 EVALUATION RESULTS

**Finding 3** *The performance of AI detectors experiences a decline on the real dataset SAID, compared to the synthetic datasets.*

The evaluation results are shown in Table 5. The results reveal a discernible decrement in performance for all the detectors on the SAID dataset compared to the synthetic dataset HC3 (Guo et al., 2023). The performance of HelloSimpleAI and MPU noticeably declined. On the HC3 dataset, their F1 scores were 96.4% and 98.4%. However, on the SAID-Quora, their F1 scores experienced a respective decrease of over 14%. These findings underline the inherent limitations of detectors

that are traditionally trained and optimized on synthetic datasets when confronted with the divergent statistical properties of AI-generated content encountered in real-world scenarios. Consequently, there's a need to recalibrate and refine the AI detectors to acclimate to ensure their relevancy and robustness in practical applications.

Table 5: Performance of Different AI Detectors on SAID and HC3.

|  | SAID-Zhihu | | | | SAID-Quora | | | | HC3 |
| --- | --- | --- | --- | --- | --- | --- | --- | --- | --- |
|  | accu. | prec. | recall | f1 | accu. | prec. | recall | f1 | f1 |
| HelloSimpleAI | 96.1 | 97.6 | 95.8 | 96.7 (**+0.3%**) | 84.4 | 84.1 | 79.8 | 81.9 (**-14.5%**) | 96.4 |
| GPTZero | - | - | - | - | 89.1 | 97.0 | 81.3 | 88.5 | - |
| MPU | 93.9 | 97.1 | 92.3 | 94.7 (**-3.7%**) | 89.3 | 96.0 | 74.7 | 84.0 (**-14.4%**) | 98.4 |

## 7 USER-ORIENTED AI-GENERATED TEXT DETECTION

In addition to the normal experiments and analyses, we introduce a novel challenge in AI-generated text detection—identifying whether responses are AI-generated based on both the content of the response and user information. We term this approach *user-oriented AI-generated text detection*. Here, user information encompasses the user's other responses, making this task more practical and effective.

### 7.1 RATIONALE

The conventional methods predominantly focus on analyzing the textual content alone . However, AI users tend to disseminate a multitude of AI-generated texts, making the user's other responses a rich and untapped source of valuable information for AI-generated text detection. Thus, leveraging user information can significantly enhance the detection capabilities and improve the accuracy of discerning current responses.

### 7.2 TASK SETUP

For this user-oriented detection, the input features include both the response content and corresponding user information, which includes their other responses. The goal is to classify responses as AI-generated or human-generated based on this amalgamated information.

**Task Formulation:** Our training data is structured as $(x_0, u(x_0), y)$, where $x_0$ represents the target response, $u(x_0) = \{x_1, \ldots, x_n\}$ denotes all $n$ responses from the user corresponding to $x_0$, and $y \in \{\text{AI-generated}, \text{human-generated}\}$ is the response label. We aim to learn a model $f(x, u(x))$ that maximizes its accuracy on the test set. This formulation allows the model to incorporate the collective responses from a user, thereby enhancing the classification accuracy of individual responses. The models are assessed using accuracy, precision, recall, and F1-score.

**Dataset:** We utilize the SAID dataset, extracting relevant user information and response content. For each data split, we randomly select 50 AI users and 50 human users, utilizing their responses as the training set. Similarly, we randomly choose 100 AI users and 100 human users to compose the test set with their responses. To avoid data leakage, we ensure that there is no overlap of users between the training and test sets.

**Detector Architecture:** To demonstrate the efficacy of the task, we design detectors built upon existing detectors. We employ the detection results of existing detectors on individual responses as model features. For an input $(x_0, u(x_0) = \{x_1, \ldots, x_n\})$, these responses are independently analyzed by the existing detectors. We use their detection results as input features $d(x_0), d(x_1), \ldots, d(x_n)$, where $d(x_i)$ represents the detection result of $x_i$ ($d(x_i) \in [0-1]$), with higher scores indicating higher likelihood of being AI-generated. Based on these features, we have designed the following three detectors:

- **Max Pooling:** This computes the maximum value of all $d(x_i)$, i.e., $max_i d(x_i)$, as the prediction result and utilizes 0.5 as the threshold for classification. The rationale behind

Table 6: Accuracy Comparison (%)

| | HelloSimpleAI | | GPTZero | | MPU | |
|---|---|---|---|---|---|---|
| | Zhihu | Quora | Zhihu | Quora | Zhihu | Quora |
| original | 96.1 | 84.4 | - | 89.1 | 93.9 | 89.3 |
| max | 97.8 | 76.1 | - | 92.2 | 95.7 | 90.8 |
| mean | **98.6 (+2.5)** | **88.2 (+3.8)** | - | 92.8 | 97.1 | 90.3 |
| MLP | 90.4 | 83.7 | - | **95.3 (+6.2)** | **98.2 (+4.3)** | **92.5 (+3.2)** |

this approach is that if any response is AI-generated, all other responses from that user are likely to be AI-generated.

- **Mean Pooling:** This calculates the average value of all $d(x_i)$, i.e., $mean_i d(x_i)$, as the prediction result and employs 0.5 as the classification threshold.

- **MLP Detector:** This is a one-layer perceptron classifier, using $d(x_i)$ as input and outputting a value between 0 and 1, representing the likelihood of the response being AI-generated. In this paper, we set the dimension of the hidden layer to 256.

It's noteworthy that both max pooling and mean pooling classifiers are training-free. We can directly leverage existing detectors for the two architectures.

**Formats of Detection Result:** Both HelloSimpleAI and GPTZero yield the probability of the target text being identified as AI-generated, denoted as $d(x_i)$. MPU, on the other hand, only produces labels for the target text. We convert these labels into 0 (human-generated) and 1 (AI-generated) to represent $d(x_i)$.

### 7.3 RESULTS

**Finding 4** *User-oriented AI-generated text detection can significantly enhance detection effectiveness.*

In Table 6, our empirical findings underline the proficiency of user-oriented AI-generated text detection, setting a benchmark in tackling the evolving challenges of realistic detection. The three detector architectures—mean, max, and MLP, exhibit superior performance in most cases.

Even without necessitating any additional training, the strategies incorporating the utilization of maximum and mean values have demonstrated to outperform the original models by a considerable margin. The mean strategy brings enhancements across all the evaluated scenarios.

MLP model has also showcased superior efficacy, especially for GPTZero, where it enhances the detection accuracy by +6.2%, and for MPU, contributing to an improvement of +4.3% and +3.2%. These substantial advancements underscore the capability of training a more sophisticated classifier.

## 8 CONCLUSION

This paper introduced SAID, a benchmark aimed at enhancing the realism and diversity in evaluating AI-generated text detectors, utilizing actual content from Zhihu and Quora. Our findings contradict prevailing assumptions, revealing that individuals adept with LLMs and social media can accurately differentiate AI-generated from human-generated responses, with a notable 96.5% accuracy rate.

We also introduced a practical and effective user-oriented AI-generated text detection challenge, emphasizing the relevance of user information and content analysis for detection, substantiated by our experimental results.

Finally, the empirical evaluation conducted on contemporary detectors using the SAID benchmark illustrated their limitations and capabilities in real-world settings, pointing towards the need for more refined and resilient detection methods in the evolving landscape of AI-generated content. This work is anticipated to spur advancements and innovations in the field, aiding the development of more reliable and robust detection mechanisms in the proliferating era of AI-generated text.

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
