# OpenReview forum: "Who SAID that? Benchmarking Social Media AI Detection"
_ICLR.cc/2024/Conference — ICLR 2024 Conference Withdrawn Submission_

### Official Review · Reviewer_mwr2 · 2023-10-20

**Soundness:** 2 fair
**Presentation:** 3 good
**Contribution:** 2 fair
**Rating:** 3
**Confidence:** 4

**Summary:**

The paper presents SAID, a benchmark to assess the AI-generated text that is shared on two online platforms: Quora and Zhihu. The paper releases a large-scale dataset of English and Chinese posts that are either AI-generated or human-generated (based on labels or platform features from Quora and Zhihu). Also, the paper investigates whether humans can identify AI-generated text, what is the performance of existing detectors in identifying AI-generated text, as well as designing a classification task that considers both content and user information to classify content as AI-generated content or human-generated content.

**Strengths:**

The paper focuses on an important and timely issue that I believe will become even more important in the future, so these research efforts and benchmarks can equip the community with the necessary tools to being able to identify machine-generated text. An important strength of the paper is that it releases a large-scale dataset of AI and human-generated posts shared on two social media platforms across two languages (English and Chinese). The multilingual and multi-platform aspect of the released dataset is worth noting, given that it likely increases the probability of being used by other researchers focusing on similar topics. Finally, I find the idea of incorporating both content information and user information to identify AI-generated content interesting and it provides a new perspective related to AI-generated content that is particularly shared on social media platforms like Quora and Zhihu.

**Weaknesses:**

I think that the paper has some considerable flaws regarding the methods/assumptions made by the paper, the interpretation of the results, the lack of validation of the employed methods, and the lack of scope. I elaborate on my main concerns along with some suggestions on how to improve the manuscript below.

**Validation and Assumptions:** I think that the paper uses methods for identifying machine-generated text that are not properly validated. Specifically, the paper uses labels provided by Zhihu and the collapsing feature in Quora to determine if the text is machine-generated or not. The issue is that these methods are essentially black boxes, and we do not know how accurate these methods are in determining whether text is machine-generated or not. Also, we do not have any clue how generalizable the methods are and how they can get deceived depending on the prompt that a user will use to generate text (e.g., a user leveraging a prompt that aims to generate text that looks like it is written by a human). On top of this lack of validation of the methods, the paper makes an assumption that any text posted by users that have many posts with these labels are actually generated by machines. Again this is an assumption that is not validated by the paper and it is unclear how the lack of the validation of methods and the assumptions made are likely to affect the quality of the released dataset. A potential way to mitigate these issues is to make a manual validation of the a small sample of the dataset to verify how many posts are indeed machine-generated based on the perceptions of the manual annotators. This help us understand the quality of the dataset and shed light into the potential false positives that exist in the released dataset. This is of paramount importance as the paper aims to release a benchmark dataset that will be used by other researchers, hence the quality of the dataset is very important.

**Sample sizes and interpretation of the results:** Another important concern with the paper is that the results presented in Section 4 are based on the perspectives of two participants only, hence the results do not provide much confidence. Also, the interpretation of the results and the comparisons with the OpenAI work is not fair given that the previous work used non-expert users, whereas here, you use participants who are familiar with LLMs and social media. Overall, I suggest to the authors to provide more details on the recruited participants (e.g., are participants authors of this paper?) and extend the annotation process so that it includes more annotators. I believe that making generalizable conclusions based on two participants is not ideal to say the least. Also, I suggest rephrasing the interpretation of these results and comparisons with previous work to reflect that here you are using expert participants while previous work did not use expert participants (which might likely be one of the reasons for the observed differences in the accuracy). Also, I am a little bit puzzled on why the paper elected to use samples and not the entire dataset for the analysis in Section 6. Using the entire dataset will give a more comprehensive view of the dataset and provide more holistic results. I suggest either using the entire dataset or justifying the need to sample the dataset.

**Scope and Focus of the Paper:** Overall, I believe that the paper tries to study too many aspects of the problem, which results in undertaking a very shallow and not validated analysis of multiple aspects of the problem. I suggest to the authors to limit the scope of the paper and focus on some of the aspects and look at them deeper and ensure that the employed methods are validated and can provide confident results. For instance, I believe that as the paper is, Section 4 is very shallow and does not provide confident results.

Overall, given the above-mentioned concerns, I believe that the paper is not ready for publication yet. I encourage the authors to continue working on this important topic and improve their manuscript and the released dataset.

**Questions:**

1. How accurate are the Zhihu labels and how robust are to machine-generated text that aims to avoid detection (based on the user prompt)?
2. How accurate is the detection of machine-generated text when using the collapse feature in Quora?
3. What is the prevalence of false positives that exist in the released dataset (i..e, text that is considered machine generated when it’s not simply because of the used methods or the user-based assumptions made)?

---

### Official Review · Reviewer_epQ3 · 2023-10-28

**Soundness:** 2 fair
**Presentation:** 3 good
**Contribution:** 2 fair
**Rating:** 5
**Confidence:** 3

**Summary:**

This paper studies an interesting problem: how to detect AI-generated contents on real social media environment. The authors collected dataset from Quara and Zhihu, two big question-answering social media platforms. As the raw data is labeled with the two platforms' API, whose number is limited, the authors applied a label expansion strategy to acquire more data. With the expanded data, the authors conducted a series of expeirments to evaluate the detection ability of human verifiers and detection models.

**Strengths:**

1. This research problem is interesting

2. The writing of this paper is easy to follow

**Weaknesses:**

My main concerns are on the data collection method of this paper. I am not convinced by the experiments. The reasons are as follows:

1. The reliability of the original label pool: The initial label pool is from the API of Quora and Zhihu. But their detection ability is not supposed to be better than the SOTA detectors and human verifiers, as they do not have more knowledge about this task. Actually, it is very possible that their initial labels are just from SOTA detectors and human verifiers. This is supported by the high accuracy of SOTA detectors and human verifiers in the experiment section. The overall lower accuracy on Quora compared to Zhihu is another evidence: The Zhihu data are all those samples that they detected as "Suspected AI-generated", thus it aligns better with the result of SOTA detectors, compared to Quora data which contains some other kinds of data violating the rules.

2. Social media platforms usually tend to apply more cautious strategy to detect the AI-generated contents. Thus, only those samples with very significant evidence will be labeled.

3. The experiment of using user profile to boost the detection does not make sense. This is because when using the label expansion strategy, the authors have already used the user identity information (they label the post as AI-generated when the three out of ten posts from the posting user is labeled by the API). Thus, the labeling algorithm has already encoded the user information into the labels. Then the user profile will naturally boost the detection when we use such labels to do evalutation.

4. The features in Table 2 are not very convincing for me. It seems that as long as the response is very rational, objective, neutral and well organized, then the labeling algorithm believes that it looks more like AI-generated. Although AI-generated contents do tend more to be in such a style, but high-quality contents written by professional experts also tend to be in such a style.

**Questions:**

1. How is the threshold of ``three out of the first ten" decided?

2. Why is the ``remind" in Table 2 considered as evidence?

---

### Official Review · Reviewer_MEvR · 2023-11-06

**Soundness:** 2 fair
**Presentation:** 3 good
**Contribution:** 2 fair
**Rating:** 5
**Confidence:** 4

**Summary:**

The paper discusses the data collection of AI and human generated content from two websites (English and Chinese). They then analyse the accuracy of humans and three AI classifiers and train various models on the output of the classifiers to improve the classification.

**Strengths:**

- The number of collected comments is large
- The methods are evaluated on two languages
- Human verification is also used
- Improvement of the classification is achieved

**Weaknesses:**

Unfortunately, I see a major issue with the data collection.

1. The classification of AI content is mainly based on the pre-labelled data from one of the websites, which means, that their classifier is defined as ground truth. We do not know anything about the used classifier. It could be one of the three classifiers used in the paper, it could also be one specifically trained on the content of the webiste

2. If a user was labelled as AI content for three of their posts, all content was labelled as AI content. This is not further justified. And it is not clear, if this is a good approach.

3. The data is unbalanced. There is far more AI content compared to human content (30% for Zhihu, and approx 60% for Quora).

4. The collected data are all answers to posts, not original posts. Yet, the conclusions are drawn for general AI-generated texts.

It is stated, that humans have a high accuracy on the given data set, which could be, that AI-generated text is easier to distinguish when it is an answer to a question (as the examples in the paper show).

**Questions:**

- Please elaborate more on the captions of each table and figure.
- Finding 3 states, that the performance of the classifiers decline on your data set, compared to the synthetic data set. What are the differences between the two data sets, that could explain the decline?